# Recent Advances in Proteinuric Kidney Disease/Nephrotic Syndrome: Lessons from Knockout/Transgenic Mouse Models

**DOI:** 10.3390/biomedicines11071803

**Published:** 2023-06-23

**Authors:** Ryosuke Saiki, Kan Katayama, Kaoru Dohi

**Affiliations:** Department of Cardiology and Nephrology, Mie University Graduate School of Medicine, Tsu 514-8507, Japan; ryosuke-s@med.mie-u.ac.jp (R.S.); dohik@med.mie-u.ac.jp (K.D.)

**Keywords:** genetic kidney disease, knockout, nephrotic syndrome, proteinuria, transgenic

## Abstract

Proteinuria is known to be associated with all-cause and cardiovascular mortality, and nephrotic syndrome is defined by the level of proteinuria and hypoalbuminemia. With advances in medicine, new causative genes for genetic kidney diseases are being discovered increasingly frequently. We reviewed articles on proteinuria/nephrotic syndrome, focal segmental glomerulosclerosis, membranous nephropathy, diabetic kidney disease/nephropathy, hypertension/nephrosclerosis, Alport syndrome, and rare diseases, which have been studied in mouse models. Significant progress has been made in understanding the genetics and pathophysiology of kidney diseases thanks to advances in science, but research in this area is ongoing. In the future, genetic analyses of patients with proteinuric kidney disease/nephrotic syndrome may ultimately lead to personalized treatment options.

## 1. Introduction

Nephrotic syndrome (NS) is defined by proteinuria and hypoalbuminemia associated with edema and hyperlipidemia. General edema and pleural effusion are observed in severe cases, and NS is a cause of end-stage kidney disease (ESKD). Even slight proteinuria affects all-cause and cardiovascular mortality [1]. NS consists of steroid-sensitive NS (SSNS) and steroid-resistant NS (SRNS). A previous report suggested that approximately 30% of SRNS cases were caused by a single gene abnormality [2]. Hereditary types of NS that appear in both familial and nonfamilial patients and manifest throughout a wide age range and spectrum of histological abnormalities have been linked to numerous genes. To name a few of the more famous ones, podocyte-specific genes include *NPHS1*, *NPHS2*, *WT-1*, *PLCE1*, *LMX1B*, *SMARCAL1*, *COQ2*, *CD2AP*, *ACTN4*, *TRPC6*, and *INF2*. As glomerular basement membrane (GBM) components, *COL4A3*, *COL4A4*, *COL4A5*, and *LAMB2* have been identified as major causes [3]. The GBM is an essential component of the glomerular filtration barrier and is also related to proteinuria [4]. For example, a loss of GBM heparan sulfate chains, major components of the GBM, is associated with proteinuria in several glomerular diseases, including lupus nephritis and diabetic nephropathy [4,5,6]. Moreover, mature podocytes produce high levels of VEGF-A, and adult glomerular endothelial cells (GECs) express high levels of VEGF receptors, which indicates that podocytes and GECs are related to each other [7]. For example, *RRM2B* deletion shows the marked endothelial hypertrophy in addition to progressive podocyte hypertrophy [8]. GEC dysfunction is characterized by a compromised endothelial glycocalyx, an inflammatory phenotype, mitochondrial damage and oxidative stress, aberrant cell signaling, and endothelial-to-mesenchymal transition in the early stages of focal segmental glomerular sclerosis (FSGS) and diabetic kidney disease (DKD) [9]. The endothelium may play a role in some glomerular diseases because it involves alterations of the systemic and glomerular endothelium and glycocalyx, and their sera directly activate GECs [10]. Glomerular endothelial cell failure is sufficient to promote podocyte damage, proteinuria, and mesangial cell activation [9]. With advances in medicine, new causative genes for genetic kidney diseases are being discovered increasingly frequently.

In this review, we summarized the findings of articles with causative genes that have been demonstrated in mouse models with proteinuria or NS, focusing on articles published within the last five years. 

## 2. Methods

We conducted a search of PubMed using the following search formula: ((“nephrotic syndrome”[All Fields] OR (“proteinuria”[MeSH Terms] OR “proteinuria”[All Fields] OR “proteinurias”[All Fields])) AND (“genes”[MeSH Terms] OR “genes”[All Fields] OR “gene”[All Fields]) AND (“mice”[MeSH Terms] OR “mice”[All Fields]) NOT “review”[Publication Type]) AND (y_5[Filter])). A total of 366 articles were identified using this search strategy. We checked all the titles and abstracts of the articles to identify pertinent articles. Following the abstracts, the entire content of the remaining papers was examined to determine their applicability to this investigation. Specifically, previous studies that were not connected to proteinuria or nephrotic syndrome, or that did not include a mouse model, were eliminated. Finally, we selected 80 articles.

## 3. Proteinuria/NS

Many genes that are related to proteinuria and NS have been discovered so far, and an overview of the genes is shown in Figure 1. 

Absent in melanoma-2 (AIM2) is an innate immune sensor for cytosolic dsDNA and localizes to podocytes in the kidney [11]. *Aim2*^−/−^ mice showed crescent formation after intravenous administration of nephrotoxic serum compared to wild-type (WT) mice, which was associated with podocyte dedifferentiation and parietal epithelial cell activation (Table 1) [11].

BMAL1 is the protein at the core of the circadian clock and regulates the transcription of various clock-controlled genes [12]. Loss of Bmal1 in the kidney can result in various alterations in the renal physiological function, such as lowering blood pressure, disrupting fluid–electrolyte balance, and changing the glomerular filtration rate [13]. The urinary albumin/creatinine ratio, serum creatinine, and blood urea nitrogen values were elevated in proximal tubular cell-specific *Bmal1* knockout mice fed an adenine diet compared to WT mice (Table 1) [14].

CRIF1 plays an essential role in mitochondrial synthesis and membrane integration of oxidative phosphorylation polypeptides, interacting with proteins surrounding the polypeptide exit tunnel of the large subunit of mitochondrial ribosomes [15]. Podocyte-specific Crif1 knockout mice exhibited progressive albuminuria and kidney dysfunction (Table 1) [16]. Electron microscopic analyses demonstrated mitochondrial structural abnormalities, such as abnormal arrangement and loss of cristae as well as podocyte foot process effacement [16].

The exocyst complex comprises eight proteins that have been shown to play vital roles in exocytosis and vesicle trafficking [17]. Two patients with deletions in *EXOC4* were identified among 256 patients with NS [15]. Podocyte-specific *Exoc5* knockout mice showed massive proteinuria, foot process effacement, and loss of slit diaphragm as well as mislocalization of nephrin and Neph1 (Table 1) [18].

GALNT11 encodes a member of the large glycosyltransferase family responsible for initiating mucin-type O-glycosylation of secreted and membrane-bound proteins [19]. Galnt11 was specifically expressed in the mouse proximal tubules similar to expression patterns seen in human kidneys, and Galnt11 knockout mice displayed increased albumin-to-creatinine ratios relative to controls, suggesting that Galnt11 had an effect on reabsorbing albumin in the proximal tubules (Table 1) [20].

Glycogen synthase kinase-3 (GSK3) has two isoforms, GSK3α and GSK3β, and plays a role in phosphorylation, protein complex formation, and subcellular distribution [21]. Podocyte-specific GSK3 α/β knockout mice (podCreGSK3α^fl/fl^β^fl/fl^) had enlarged pale kidneys, kidney failure, and high levels of albuminuria (Table 1) [22]. Podocin RtTA-tet-o-Cre GSK3α^fl/fl^β^fl/fl^ mice given doxycycline from four weeks old developed a spectrum of kidney disease, ranging from mild albuminuria or mesangial hypercellularity to glomerulosclerosis and interstitial fibrosis [22]. These data showed essential roles of GSK3 α/β in the developmental and maturity periods [22].

Although many causative genes with SRNS have recently been discovered, not much is known about the genes associated with SSNS, which accounts for approximately 80% of childhood-onset NS [23]. Analyzing a case of familial SSNS indicated the potential causative gene to be IL1RAP, which encoded an essential common subunit of the functional IL-1, IL-33, and IL-36 receptors [24]. Peripheral blood mononuclear cells in SSNS patients showed a decreased response to IL-1β. Furthermore, *Il1rap* knockout mice exhibited exacerbated lipopolysaccharide (LPS)-induced nephrotic albuminuria (Table 1) [24].

Mutations in six genes—*MAGI2*, *TNS2*, *DLC1*, *CDK20*, *ITSN1*, and *ITSN2*—were found in seventeen families with NS [25]. Although there were no marked differences between *Itsn2*^L−/L−^ and WT mice in the histologic findings of the kidneys or level of urinary protein, LPS injection increased urine albumin levels in *Itsn2*^L−/L−^ mice compared to WT mice (Table 1) [25]. There was also a delayed recovery from podocyte injury in *Itsn2*^L−/L−^ mice compared to WT mice [25].

The LAMA5 gene encodes Laminin-α5, and *Lama5*^−/−^ mice exhibited embryonic lethality and severe defects in glomerular development (Table 1) [26]. Three hundred families with pediatric NS underwent whole-exome sequencing, which revealed homozygous variants in *LAMA5* in three families [27]. These genetic variants might contribute to the development of NS in pediatric patients.

PALLD plays a critical role in the stability and dynamics of the actin cytoskeleton [28]. Podocyte-specific Palld knockout showed disrupted morphology of the glomeruli with mild podocyte foot process effacement and increased susceptibility to nephrotoxic serum (Table 1) [28].

Podocyte-specific *PP2A* knockout mice exhibited weight loss, growth retardation, and proteinuria associated with progressive glomerulosclerosis, interstitial fibrosis, and foot process effacement (Table 1) [29]. Upregulation of phosphorylated YB-1 was observed in podocyte-specific *PP2A* knockout mice and might be related to the functional integrity of glomerular filtration [29].

Repressor element 1-silencing transcription factor (REST) is a repressor of neuronal genes during embryonic development [30]. Although REST is downregulated after terminal neuronal differentiation, it is induced in the aging human brain and regulates a network of genes that mediate cell death and stress resistance [30]. Podocyte-specific *Rest* knockout mice developed albuminuria, glomerulosclerosis, and interstitial fibrosis (Table 1) [31]. Furthermore, REST was found to be induced by oxidative stress and protected against apoptosis in podocytes [31].

Shroom3 is an F-actin binding protein that is important for epithelial morphogenesis [32]. Although tubular-specific Shroom3 knockdown in mice inhibited kidney fibrosis in a ureteric obstruction model [32], glomerular and podocyte-specific Shroom3 knockdown induced reversible albuminuria with podocyte foot process effacement without podocyte loss (Table 1) [32,33].

Ste20-like kinase (SLK) is a serine/threonine kinase expressed ubiquitously and appears to be a regulator of cytoskeletal structure [34]. Podocyte-specific *Slk* knockout mice injected with adriamycin showed a decreased number of podocytes and greater albuminuria than control mice (Table 1) [34]. Ezrin levels and ezrin phosphorylation were reduced in podocyte-specific *Slk* knockout mice injected with adriamycin, which was associated with the decreased expression of F-actin and alteration of the shape of podocytes [34].

Twist1 is a transcriptional repressor and inhibits cytokine production by diminishing NF-κB or Runx3 expression in Th1 cells [35]. Twist1 in podocytes limited CCL2 production and macrophage infiltration in injured glomeruli [36]. Although podocyte-specific Twist1 knockout mice did not exhibit proteinuria, they had more proteinuria than WT mice after inducing nephrotoxic serum or adriamycin (Table 1) [36]. 

The degradation systems for cellular proteins consist of the ubiquitin proteasome system and the autophagosome–lysosomal pathway, central to which is the conjugation of ubiquitin to substrate proteins [37]. Deubiquitinating enzymes (DUBs) make ubiquitination reversible, slowing the ubiquitination process by removing ubiquitin chains or inhibiting the catalytic function of ubiquitin-related enzymes [38]. Ubiquitin-specific protease (USP) is the largest family of DUBs, and USP40 is specifically localized in the podocytes of the mature glomerulus [38]. Cultured podocytes with *USP40* knockdown decreased HINT1 and p53 [39]. Although Usp40 knockout mice did not exhibit any alterations in the glomerular phenotype, USP40 and its interacting partners formed a regulatory network that protected the cellular processes leading to glomerular sclerosis (Table 1) [39].

WTIP is part of a multiprotein complex in the podocyte foot process and shuttles between the nucleus and cytosol [40]. Wtip^−/−^ mice exhibited embryonic lethality, and *Wtip* heterozygous mice developed significant proteinuria in response to LPS or adriamycin injection compared to WT mice (Table 1) [41]. Further studies involving podocyte-specific *Wtip* knockout mice are desirable.

## 4. Focal Segmental Glomerular Sclerosis (FSGS)

FSGS is the most common glomerular histologic lesion associated with high-grade proteinuria and ESKD, which can be caused by a variety of underlying mechanisms [42]. In individuals who either do not receive treatment or are refractory to it, primary FSGS is often a progressive condition, with a 5% rate of spontaneous remission and a 50% rate of ESKD during a period of 5–8 years following a biopsy [42]. Up to two-thirds of patients with FSGS who present in the first year of life have genetic abnormalities that account for the later clinical presentation in this age range [42]. However, in older children and adults with FSGS and a related genetic mutation, the direct causal relationship with the disease process, such as proteinuria and kidney failure, is not as clear. In this situation, it has been proposed that a second hit might be necessary [42]. There are many theories as to where these triggers come from, including additional genetic and/or outside environmental elements [42]. Further investigations into causal genes might provide more information on treatment efficacy and the kidney prognosis.

Polymorphisms in APOL1 are a risk factor for chronic kidney disease (CKD), including human immunodeficiency virus (HIV)-associated nephropathy and FSGS [43]. The common allele (known as G0) reduced glomerulosclerosis in a murine model of HIV-associated nephropathy compared to the CKD-associated risk alleles, known as variants G1 and G2 [43]. The APOL1 G1 risk allele made mice more susceptible to kidney disease in a lipotoxicity-driven FSGS model (Table 2) [44]. Intravenous injection of interferon γ led to heavy proteinuria and glomerulosclerosis in G1/G1 and G2/G2 but not G0/G0 bacterial artificial chromosome (BAC) transgenic mice [45]. Transmission electron microscopic analyses of human urinary podocytes showed a reduced mitochondrial matrix density and increased mitochondrial area in G1/G2 podocytes compared with G0/G0 podocytes [44]. Antisense oligonucleotides against APOL1 mRNA reduced protection against IFN-γ–induced proteinuria in APOL1 G1 mice [46]. In humans with two APOL1 variants (G1/G1, G2/G2 or G1/G2), inaxaplin selectively inhibited the APOL1 channel function and reduced proteinuria in a phase 2a study [47]. Further studies will be expected in the future.

Rho GTPases, such as RhoA, Rac1, and Cdc42, are regulators of the actin cytoskeleton and play important roles in podocyte morphology and ensuring an efficient barrier function [48]. Arhgef7 is an important activator of Cdc42, and podocyte-specific *Arhgef7* knockout mice exhibited progressive proteinuria and FSGS with reduced Cdc42 activity (Table 2) [49].

COQ6 is needed for the biosynthesis of coenzyme Q10, and its mutations in human patients produce NS with sensorineural deafness [50]. Recently, a new mutation of *COQ6* c.41G>A was detected in a patient with FSGS [51]. Podocyte-specific *Coq6* knockout mice developed proteinuria and FSGS (Table 2) [52]. However, podocyte-specific *Coq6* knockout mice treated with 2,4-dihydroxybenzoic acid, which functioned to bypass certain deficiencies of the CoQ10 biosynthesis pathway, were protected from kidney disease progression, showing an improved survival compared to untreated mice [52].

Crb2 is a type I transmembrane protein that is expressed in the apical membrane of podocytes [53]. Podocyte-specific Crb2 knockout mice had severe albuminuria, FSGS, and tubulointerstitial fibrosis associated with a decreased expression of Nphs2, Podxl, and Nphs1 (Table 2) [53]. CRB2 protein variants with SRNS accumulated in the endoplasmic reticulum (ER), exhibited altered glycosylation patterns, and induced an ER stress response [54].

MYO9A has a Rho-guanosine triphosphatase activating protein (Rho-GAP) tail domain that deactivates RhoA [55]. *Myo9a*^R701X/+^ mice showed proteinuria and FSGS with increased RhoA activity, which recapitulated autosomal dominant inheritance of the heterozygous MYO9A p. R701X variant identified in the proband (Table 2) [55].

PARVA controls RhoA/ROCK-mediated contractility [56]. Podocyte-specific Parva knockout mice exhibited proteinuria and FSGS, which resulted in kidney dysfunction (Table 2) [57,58]. PARVA associated with TJP1 (also known as ZO-1) and prevented lysosome-dependent degradation of TJP1, which contributed to maintaining the podocyte structure and function [57]. Podocyte-specific *Tjp1* knockout mice showed proteinuria and GS with impaired slit diaphragm formation; in addition, podocyte-specific Tjp1 and *Tjp2* double-knockout mice showed the accelerated appearance of the defects observed in podocyte-specific *Tjp1* knockout mice [59].

Urokinase-type plasminogen activator receptor (uPAR) is a glycosyl-phosphatidylinositol (GPI)-anchored protein, and soluble uPAR (suPAR) is generated by removal of the GPI anchor from uPAR [48]. suPAR can be detected in blood and urine and serves as both an inflammatory biomarker and a signaling molecule [60]. suPAR isoform-2 transgenic mice developed albuminuria and FSGS with podocyte foot process effacement (Table 2) [61].

Phosphatidylserine is asymmetrically and dynamically distributed across the lipid bilayer in eukaryotic cell membranes, which is maintained by flippases, one of the most important P4-ATPases [62]. The TMEM30 (also known as CDC50) family proteins interact with multiple P4-ATPases [63], and the TMEM30A expression was shown to be decreased in patients with minimal change disease and membranous nephropathy (MN) [64]. Podocyte-specific *Tmem30a* knockout mice showed albuminuria and FSGS, which was associated with ER stress (Table 2) [64].

Zinc finger and homeobox (ZHX) family transcription factors, such as ZHX1, ZHX2 and ZHX3, regulate the majority of structurally and functionally important podocyte genes, and ZHX2 is one of the most potent transcriptional repressors of WT1 [65]. While podocyte-specific Zhx2 knockout mice did not show albuminuria compared to control mice, podocyte-specific Zhx2 transgenic rats showed more proteinuria than WT rats after adriamycin injection (Table 2) [65].

## 5. MN (Membranous Nephropathy)

MN is a kidney glomerular condition that is diagnosed pathologically and is characterized by thickening of the glomerular capillary walls caused by the development of immune complexes on the outer portion of the basement membrane [66]. MN accounts for 30% of adult cases of nephrotic syndrome. In 80% of MN patients, there is no underlying etiology of MN, while 20% of cases are associated with drugs, such as nonsteroidal anti-inflammatory drugs, or other disorders, such as systemic lupus erythematosus, hepatitis B or C, and malignancies [66]. Although MN is not a typical Mendelian hereditary disease, emerging evidence suggests a significant genetic component [66].

A 40-year-old woman with Charcot–Marie–Tooth disease developed nephrotic range proteinuria due to MN, and a genetic analysis identified a heterozygous nonsense variation in exon 2 of the MPZ gene [67]. MPZ is an integral membrane glycoprotein and is essential for membrane adhesion [68]. Mutations in *MPZ* were associated with Charcot–Marie–Tooth disease, and *Mpz*^−/−^ mice exhibited higher rates of albuminuria and GBM thickening than WT mice (Table 2) [68].

NPNT is an extracellular protein localized in the GBM, and injection of miR-378a-3p, which targets NPNT, led to albuminuria and podocyte foot process effacement in mice [69]. Podocyte-specific Npnt knockout mice showed proteinuria and widening of the lamina rara interna of the GBM (Table 2) [70].

## 6. Diabetic Kidney Disease (DKD)/Diabetic Nephropathy (DN)

Glomerular hypertension, altered renal hemodynamics, ischemia and hypoxia, oxidative stress, and activation of the renin–aldosterone pathway are all factors contributing to the etiology of DKD [71]. The “metabolic memory” phenomenon is a key factor in the development of DKD [71]. Even after receiving therapy, patients who have previously experienced hyperglycemia have been shown to continue to experience problems, such as DKD [71]. Further genetic research is, thus, necessary, as these processes have the potential to lead to innovative therapies for DKD.

G protein-coupled receptors (GPCRs) constitute a protein family of receptors that sense molecules outside the cell and activate a number of different intracellular signal transduction pathways [72]. The expression of Gprc5a was highly specific to podocytes and was shown to be downregulated in DN [73]. Indeed, *Gprc5a*^−/−^ mice exhibited thickening of the GBM, activation of profibrotic signaling pathways, and promotion of glomerular injury in a diabetic model (Table 3) [73].

IRE1α is an ER-transmembrane protein that is activated during ER stress [74]. Podocyte-specific Ire1 knockout mice showed age-dependent albuminuria and podocyte foot process effacement (Table 3) [75]. Furthermore, podocyte-specific *Ire1* knockout mice with streptozotocin injection showed higher rates of albuminuria and accelerated DN than control mice [76].

KAT5 is a histone acetyltransferase involved in DNA damage repair, and podocyte-specific Kat5 knockout mice developed massive albuminuria and FSGS associated with increased DNA methylation of the promoter region of *Nphs1* (Table 3) [77].

PTEN is a dual-function lipid and protein phosphatase that regulates cell growth, cytoskeletal rearrangement, and motility [74]. Inducible podocyte-specific *Pten* knockout mice had much higher proteinuria than controls (Table 3) [74]. Furthermore, inducible podocyte-specific *Pten* knock-in diabetic mice exhibited ameliorated albuminuria compared to diabetic control mice, although the blood glucose levels of these two mouse groups were comparable [78]. Targeting PTEN might be a new therapeutic strategy against DKD.

RhoA expression was found to be lower in the podocytes of db/db mice, a well-known proteinuric mouse model that resembles DN, than in control mice [79]. Furthermore, albuminuria was obviously increased in murine models of *RhoA* knockdown compared to control mice (Table 3) [79].

## 7. Hypertension/Nephrosclerosis

As human life expectancy continues to increase, aging populations present a growing challenge for clinical practice [80]. Every year after the age of 30, approximately 6000 to 6500 nephrons are lost due to nephrosclerosis or glomerulosclerosis. While aging itself does not cause kidney injury, the physiological changes associated with normal aging processes are likely to compromise the kidney’s ability to repair itself, making older individuals more susceptible to acute kidney disease, chronic kidney disease, and other kidney illnesses than younger ones [80]. We discuss, therefore, a variety of signaling molecules that have been found to aggravate kidney cell senescence and kidney aging.

Autophagy was shown to be crucial for the maintenance of cellular homeostasis, particularly in podocytes, and podocyte-specific *Atg5* knockout mice showed proteinuria and glomerulopathy (Table 3) [81]. Angiotensin II-induced hypertension in podocyte-specific *Atg5* knockout mice worsened albuminuria and glomerulosclerosis, which were prevented by calpastatin overexpression [82]. Proximal tubular cell-specific *Atg5* knockout mice with a unilateral ureteric obstruction model showed impaired autophagy and cytokine production [83].

## 8. Alport Syndrome (AS)

AS was initially identified in the 1970s when distinct ultrastructural anomalies were discovered in the GBM of patients. Subsequent studies led to the identification of collagen IV as the protein responsible for AS as well as the cloning and sequencing of the *COL4A3*, *COL4A4*, and *COL4A5* genes [84]. The Alport phenotype can range from a nonprogressive kidney-limited condition to a progressive multisystem disease, with X-linked, autosomal recessive, autosomal dominant, and digenic inheritance patterns [84]. Phenotypic heterogeneity is a significant feature of disorders involving the collagen type IV α3, α4, and α5 chains, resulting in a wide range of kidney outcomes and extrarenal symptoms [84]. Genetic research may help identify the most severe cases and determine their phenotypic features.

As AS patients with the same genetic mutations exhibited a wide range of disease severities, this variability was thought to be caused partly by the existence of underlying modifier genes [85]. *Col4a5* mutant mice with *Fmn1*^+/−^ had a lower urinary albumin-creatinine ratio in both males and females than *Col4a5* mutant mice with *Fmn1*^+/+^ (Table 4) [86]. *Fmn1* is a modifier gene that mediates the severity of X-linked AS in mice by reducing the urinary albumin–creatinine ratio and reducing podocyte foot process evasion into the GBM [86].

## 9. Rare Diseases

Farber disease manifests with hoarseness and painful swollen joints accompanied by nephropathy with elevated urine ceramide levels [87]. Podocyte-specific *Asah1* knockout mice showed severe proteinuria and podocyte foot process effacement associated with ceramide accumulation in the glomeruli compared to control mice (Table 4) [88]. Increased urinary exosome excretion and impaired autophagic flux were observed due to altered TRPML1 channel activity in podocyte-specific *Asah1* knockout mice compared to control mice [89,90].

DROSHA was identified as a putative oncogenic driver of Wilms tumor [91]. Podocyte-specific *Drosha* knockout mice developed proteinuria and kidney failure without tumor formation (Table 4) [92].

Endocytosis refers to the mechanism by which cells internalize macromolecules and particles into transport vesicles derived from the plasma membrane, which is critical for the reabsorption of filtered macromolecules, such as low-molecular-weight proteins in the kidney [93]. The EHD proteins were previously implicated in endosomal scission, allowing the receptor and cargo to be separated in order to be processed at their respective proper destinations [93]. Six individuals with an unexplained unique phenotype of low-molecular-weight proteinuria and sensorineural hearing loss had a single homozygous variant in *EHD1* c.1192C>T (p. R398 W) [94]. *Ehd1*^−/−^ and *Ehd1*^R398W/R398W^ mice showed a substantial decrease in the reuptake, leading to increased urinary β_2_-microglobulin excretion (Table 4) [94].

A boy with short stature, visual impairment, and developmental delay was found to have compound heterozygous mutations in *LAMB2*, which codes laminin 2, after presenting with recurrent macroscopic hematuria and albuminuria [95]. Pierson syndrome, an autosomal recessive condition characterized by congenital NS, ocular abnormalities (commonly microcoria), muscular hypotonia, and neurological deficits, was linked to homozygous or compound heterozygous mutations in *LAMB2* [96]. *Lamb2*^−/−^ mice developed massive proteinuria (Table 4) [97].

A rare heterozygous substitution (p. Leu239Pro) in *MAFB*, a leucine zipper transcription factor, was found in two families with FSGS associated with Duane retraction syndrome, characterized by impaired horizontal eye movement as a result of cranial nerve malformation [98]. According to structural modeling, the p. Leu239Pro substitution in the DNA-binding domain might interfere with the stability of the nearby zinc finger [98]. Neonatal mice with p. Leu239Pro showed impaired differentiation in their podocytes compared to control mice (Table 4) [98].

Mitochondrial illnesses, a clinically and genetically varied category of multisystemic disorders affecting many organs, are caused by impaired replication or maintenance of mitochondrial DNA (mtDNA) [99]. Mutations in *MGME1* affect the mitochondrial function and result in mitochondrial disease syndrome [99]. *Mgme1*^−/−^ mice exhibited severe nephropathy with elevated plasma levels of urea and creatinine, leading to a shorter life span than in WT mice (Table 4) [100].

A rare recessive genetic disorder known as Galloway–Mowat syndrome causes neurodevelopmental abnormalities and progressive renal glomerulopathy, and the responsible gene is *WDR73* [101]. WDR73 depletion affected focal adhesion assembly in cultured podocytes, and podocyte-specific *Wdr73* knockout mice were found to be more susceptible to glomerular injury with adriamycin than control mice (Table 4) [102].

## 10. Additional Information

This section describes genes that have been somewhat established as causative genes for more than five years but for which new additional information has been reported.

To evaluate the pathogenicity of *NPHS2* p. R229Q, which was the most frequent missense variant, *Nphs2*^R231Q/R231Q^ mice were generated. *Nphs2*^R231Q/R231Q^ mice developed albuminuria and were more susceptible to nephrotoxic serum than control mice [103].

A new major modification locus for podocyte injury in *Tns2*^−/−^ mice was detected on chromosome 10 [104].

Sphingosine-1-phosphate lyase insufficiency syndrome, a rare metabolic condition linked with nonlysosomal sphingolipid storage, is caused by biallelic loss-of-function mutations in *SGPL1* [105]. The majority of those affected had SRNS that progressed quickly to ESKD [105]. In *Sgpl1*^−/−^ mice, SGPL1 gene transfer eliminated nephrosis, developmental delay, and lipidosis and significantly increased the survival [105].

The inflammatory process altered the WT1 expression and localization in podocytes, causing kidney injury [106]. The WT1 expression was lowest at 36 h after inflammation induction and its phosphorylated form was found in the cytoplasm mainly, which was associated with decreased *Nphs1* mRNA expression and increased tumor necrosis factor α and interleukin 1β mRNA expression [106].

The expression of THSD7A was enhanced in specific membrane domains, resulting in stabilized podocyte cell dynamics [107]. THSD7A might be involved in controlling the slit diaphragm dynamics of the glomerular filtration barrier [107].

## 11. Conclusions

To date, the functional role of many genes in podocytes has been elucidated by using podocyte-specific knockout/transgenic mouse models. There have also been reports about inflammatory mediators, mitochondria, and age-related changes in the kidney. By elucidating the abnormalities and mechanisms of genes in various parts of the kidney, kidney diseases will become clearer than they are now.

Substantial strides have been made in understanding the genetics and pathophysiology of kidney diseases thanks to advances in science, but these advances are still ongoing. In the future, genetic analyses of patients with proteinuric glomerulopathy may ultimately lead to personalized treatment.

## Figures and Tables

**Figure 1 biomedicines-11-01803-f001:**
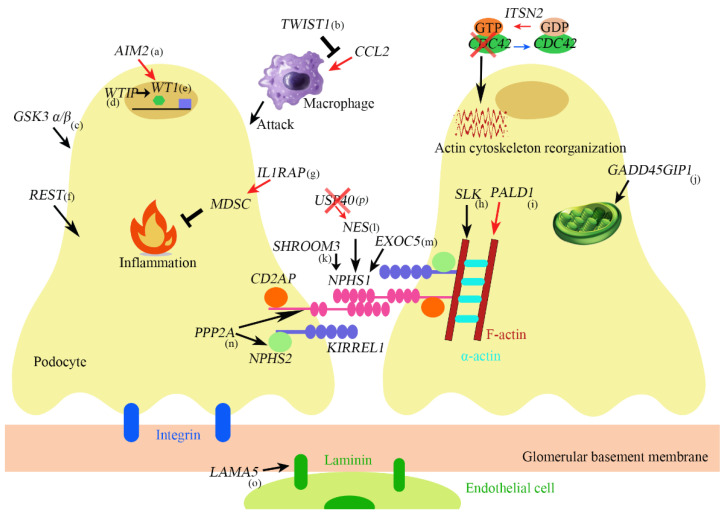
(a) AIM2 promotes podocyte differentiation and suppresses proliferation by increasing WT1 and cell cycle genes such as CDKN1A. (b) TWIST1 inhibits CCL2 induction which promotes monocyte/macrophage infiltration into the injured glomeruli. (c) GSK α/β knockout causes mitotic catastrophe. (d) If WTIP is retained in the nucleus, WTIP associates with WT1 and inhibits WT1-dependent transcriptional activation of the amphiregulin promoter. (e) WT1 is both a transcriptional repressor and activator and regulates genes important in nephron formation including podocalyxin, amphiregulin, and perhaps nephrin. (f) REST maintains cytoskeleton homeostasis protects against apoptosis and maintains kidney function during aging. (g) IL1RAP induces suppressive immune cells called myeloid-derived suppressor cells (MDSCs). (h) SLK is a regulator of cytoskeletal structure. (i) PALD1 regulates actin filaments, synaptopodin, and α-actinin-4. (j) GADD45GIP1 plays an essential role in mitochondrial synthesis and membrane integration of OXPHOS polypeptides. (k) SHROOM3 is related to Fyn activation and nephrin phosphorylation. (l) NES regulates nephrin. (m) EXOC5 affects the expression and localization of nephrin. (n) PP2A is involved in the expression of synaptopodin, podocin, nephrin. (o) LAMA5 encodes Laminin-α5. (p) USP40 is colocalized with NES in developing and mature podocytes. Its deficiency upregulates NES.

**Table 1 biomedicines-11-01803-t001:** Proteinuria-associated genes reviewed in the present article.

Gene Name	Chromosome	Protein Function	Renal Findings in the Gene Deletion
AIM2	1	An innate sensor for cytosolic dsDNA	Podocyte dedifferentiation (deletion in podocytes)
BMAL1	11	The core of the circadian clock and regulate the transcription of various clock-controlled genes	Suppressing cystathionine β-synthase transcription and expression (deletion in proximal tubule)
CRIF1	19	Mediates the integration of nascent oxidative phosphorylation polypeptides into the inner mitochondrial membrane	Mitochondrial structural abnormalities such as swelling, abnormal arrangement, and loss of cristae (deletion in podocytes)
EXOC5	14	Exocyst complexes are related to exocytosis and vesicle traffic	Foot process effacement, loss of slit diaphragm
GALNT11	7	Protease processing, secretion, cell signaling, cell adhesion, and organ growth	Glycosylates megalin and affects ligand binding
GSK3A	19	Contributing to deposition of glycogen	Fewer podocytes and reduced nephrin expression (deletion in podocytes)
GSK3B	3		
IL1RAP	3	Encodes a subunit of the functional IL-1 receptor, IL-33 receptor, and IL-36 receptor	Lower myeloid-derived suppressor cells which inhibit immune cells and ameliorate inflammation
ITSN2	2	Guanine exchange factors for Cdc42	Delayed recovery from podocyte injury
LAMA5	20	Encodes Laminin-α5 (an essential component of the GBM)	Severely disrupted glomerulogenesis, defects in the branching of the ureteric bud, and sporadic renal agenesis
PALLD	4	A pivotal role in the stability and dynamics of the actin cytoskeleton	Disrupt the morphology of podocytes (deletion in podocytes)
PPP2CA	5	Signal transmission, cytoskeleton dynamics, cell transformation, angiogenesis, etc.	Cytoskeleton rearrangement and downregulation of synaptopodin, podocin, and nephrin but not podocyte loss (deletion in podocytes)
PPP2CB	8		
REST	4	Regulates cell death and stress resistance	Maintains cytoskeleton homeostasis and protects against apoptosis during aging
SHROOM3	4	An F-actin binding protein, important for epithelial morphogenesis via Rho-kinase binding	Foot process effacement without podocyte loss (deletion in podocytes), inhibition of renal fibrosis (deletion in tubular)
SLK	10	A regulator of cytoskeletal structure	Reduce F-actin and alter the shape of glomerular epithelial cells (deletion in podocytes)
TWIST1	7	A repressor of cell-mediated and humoral adaptive immunity	Increase levels of CCL2 and TNF-α after podocytes injury (deletion in podocytes)
USP40	2	Restrain the ubiquitination process or inhibit the catalytic function of ubiquitin-related enzymes	Upregulate nestin
WTIP	19	Represses the activity of Wilms’ tumor-1 activity	Early and prolonged proteinuria in response to lipopolysaccharide

CCL2, C-C motif chemokine ligand 2; GBM, glomerular basement membrane; IL, interleukin; TNF-α, tumor necrosis factor-α.

**Table 2 biomedicines-11-01803-t002:** FSGS- and MN-associated genes reviewed in the present article.

FSGS			
Gene Name	Chromosome	Protein Function	Renal Findings in the Gene Deletion
APOL1	22	Roles in the transport and metabolism of lipids	Increase triglyceride content and alter mitochondrial structure in the podocytes (deletion in podocytes)
ARHGEF7	13	Guanine exchange factors for Cdc42	Reduces Cdc42 activity, causing podocyte apoptosis and loss
COQ6	14	Needed for the biosynthesis of coenzyme Q10	Abnormal mitochondria characterized by hyperproliferation and increased size and the defect in podocyte migration rate (deletion in podocytes)
CRB2	9	Control cell polarization and establish cell–cell contacts	A binding partner for the nephrin extracellular domain (deletion in podocytes)
MYO9A	15	Encode a nonmuscle myosin	Decreased Myo9A-actin-calmodulin interaction and increased active RhoA
PARVA	11	An adhesion checkpoint that controls RhoA/ROCK-mediated contractility	Abnormal podocyte architecture (deletion in podocytes)
PLAUR	19	This metabolic product works as an inflammatory biomarker and a signaling molecule	Activate glomerular Src kinase
TMEM30A	6	Fold and localize properly P4-ATPases into subcellular	Cause ER stress (deletion in podocytes)
ZHX2	8	Transcriptional repressors of WT1	Worse FSGS in both Zhx2 deficient and overexpressing
**MN**			
Gene name	Chromosome	Protein function	Renal findings in the gene deletion
MPZ	1	Related to membrane adhesion and compaction of the myelin membranes	Thickening of the GBM
NPNT	4	An extracellular matrix protein	Widening of the lamina rara interna of the GBM

ER, endoplasmic reticulum; FSGS, focal segmental glomerular sclerosis; GBM, glomerular basement membrane; MN, membranous nephropathy; ROCK, Rho-associated kinase.

**Table 3 biomedicines-11-01803-t003:** DKD/DN- and hypertension/nephrosclerosis-associated genes reviewed in the present article.

DKD			
Gene Name	Chromosome	Protein Function	Renal Findings in the Gene Deletion
GPRC5A	12	Sense molecules outside the cell and activate intracellular signal transduction pathways	Thickening of the GBM and activation of profibrotic signaling pathways
IRE1	17	A sensor of misfolded protein accumulation in the ER, which leads to ER stress	Relative podocyte depletion (deletion in podocytes)
KAT5	11	DNA damage repair	Increase DNA double-strand breaks and decrease nephrin expression (deletion in podocytes)
PTEN	10	Regulates a wide array of cellular processes including cell growth, migration, and metabolism	Podocyte effacement, glomerular obliteration of capillaries, and glomerular sclerosis (deletion in podocytes)
RHOA	3	Maintaining the function of cytoskeletal architecture	Podocyte apoptosis through Yes-associated protein
**Hypertension**			
Gene name	Chromosome	Protein function	Renal findings in the gene deletion
ATG5	6	ATG5-mediated autophagy suppresses inflammatory response via inhibition of NF-κB signaling	Interstitial inflammation, kidney fibrosis (deletion in proximal tubule)

ER, endoplasmic reticulum; GBM, glomerular basement membrane; NF-κB, nuclear factor-kappa B.

**Table 4 biomedicines-11-01803-t004:** AS- and rare disease-associated genes reviewed in the present article.

AS			
Gene Name	Chromosome	Protein Function	Renal Findings in the Gene Deletion
FMN1	15	Involved in actin filament and microtubule cytoskeleton formation	Foot process effacement
**Rare diseases**			
Gene name	Chromosome	Protein function	Disease
ASAH1	8	Metabolize ceramide within cells, protect podocytes from oxidative stress and apoptosis under pathologic conditions	Farber disease
DROSHA	5	An essential part of the microprocessor complex that initiates microRNA biogenesis from primary microRNAs	May cause Wilms tumor
EHD1	11	Endosomal scission	Low-molecular-weight proteinuria and sensorineural hearing loss
LAMB2	3	Mediate cell attachment, chemotaxis, and receptor binding	Pierson syndrome
MAFB	20	A leucine zipper transcription factor	Focal segmental glomerulosclerosis with Duane Retraction Syndrome
MGME1	20	Process newly replicated 5′ DNA ends to facilitate ligation when mitochondria DNA synthesis is completed	Mitochondrial disease
WDR73	15	Cell adhesion, spreading, and establishing polarity axis in cell division	Galloway–Mowat syndrome

AS, Alport syndrome.

## Data Availability

Not applicable.

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
