# Peer review of "Recent Advances in Proteinuric Kidney Disease/Nephrotic Syndrome: Lessons from Knockout/Transgenic Mouse Models"

_biomedicines, 2023, doi:10.3390/biomedicines11071803_

Round 1
Reviewer 1 Report
The paper by Saiki R. and colleagues reviewed the literature of the last 5 years on possible causative genes of proteinuria or nephrotic syndrome. The paper is overall well written and organised, however I have few comments:
- I would not limit the search to the last 5 years, since there are not many studies on the topic;
- how many articles you retrieved with your search strategy? Which databases you searched? I would add a specific Methods section;
- please define MN at its first use;
- the last column of Table 2 is confusing, with no interline between one study (row) and the other;
- although Nephrotic syndromes have historically been considered podocytes disorders, the importance of glomerular endothelial cells is now recognised in the pathogenesis of these diseases. Authors cited ref. 65 and studies on DLC1, but they do not even mention the possible role of glomerular endothelial cells dysfunction. e.g. doi: 10.1007/s00467-004-1696-5, doi: 10.1172/JCI17423, doi: 10.3389/fmed.2021.761600
- I suggest Authors to add a summarising Figure
Author Response
To Reviewer 1
The paper by Saiki R. and colleagues reviewed the literature of the last 5 years on possible causative genes of proteinuria or nephrotic syndrome. The paper is overall well written and organised, however I have few comments:
- I would not limit the search to the last 5 years, since there are not many studies on the topic;
Response
As the review articles in Biomedicines must be up to date (i.e., ≥50% are papers published within recent five years), we have attempted to search for articles within five years. As suggested, we have added sentences about major genes from articles published more than five years ago for the readers to understand the history of the topic in the Introduction section on page 1 (lines 28-34).
- how many articles you retrieved with your search strategy? Which databases you searched? I would add a specific Methods section;
Response
As suggested, we have added a specific Methods section to describe how we searched for the articles.
- please define MN at its first use;
Response
As suggested, we have defined MN at its first use at page 8, line 258.
- the last column of Table 2 is confusing, with no interline between one study (row) and the other;
Response
As suggested, we have added an interline between the end of the FSGS section and the beginning of the MN section in Table 2.
- although Nephrotic syndromes have historically been considered podocytes disorders, the importance of glomerular endothelial cells is now recognised in the pathogenesis of these diseases. Authors cited ref. 65 and studies on DLC1, but they do not even mention the possible role of glomerular endothelial cells dysfunction. e.g. doi: 10.1007/s00467-004-1696-5, doi: 10.1172/JCI17423, doi: 10.3389/fmed.2021.761600
Response
As suggested, we have added descriptions on glomerular endothelial cells by citing the suggested articles in the Introduction section on page 1 (lines 34-38).
- I suggest Authors to add a summarising Figure
Response
As suggested, we have added Figure 1 as a summarizing figure.
Reviewer 2 Report
The article raises the important issue of proteinuria and, from this point of view, is an interesting option for a review article.
However, I have doubts about the title - what mechanisms do the authors indicate that underlie kidney diseases?
The title does not reflect the content of the work.
The article is a review article and should systematize knowledge based on a table and diagrams. This article doesn't do that. I'm not sure what the purpose of such an article is.
It definitely lacks summaries; the conclusions are descriptive and do not add much.
Although you can see the enormous work of the authors, I believe that there is a definite lack of a summary, an attempt to draw, based on diagrams, how the mechanisms underlying the diseases mentioned by the authors can run.
In addition, it would be useful, for example, research perspectives...
After reading the article, I'm unsure if I learned anything crucial.
English is ok. I belive that only minor editing of English language is required.
Author Response
To Reviewer 2
The article raises the important issue of proteinuria and, from this point of view, is an interesting option for a review article.
However, I have doubts about the title - what mechanisms do the authors indicate that underlie kidney diseases?
The title does not reflect the content of the work.
Response
As suggested, we have modified the title to “Recent advances in proteinuric kidney disease/nephrotic syndrome: Lessons from knockout/transgenic mouse models”.
The article is a review article and should systematize knowledge based on a table and diagrams. This article doesn't do that. I'm not sure what the purpose of such an article is.
It definitely lacks summaries; the conclusions are descriptive and do not add much.
Although you can see the enormous work of the authors, I believe that there is a definite lack of a summary, an attempt to draw, based on diagrams, how the mechanisms underlying the diseases mentioned by the authors can run. In addition, it would be useful, for example, research perspectives... After reading the article, I'm unsure if I learned anything crucial.
Response
As suggested, we have added Figure 1 as a summarizing figure on page 2 and research prospects in the Conclusions section on page 12 (lines 417-421).
Round 2
Reviewer 1 Report
The manuscript has significantly improved, however I still have some concerns:
- Authors keep understating the role of GBM and glomerular endothelial cells. Few sentences in the introduction section are not sufficient to cover the topic, which is an important and emergent topic in nephrotic syndromes research. As previously suggested, I recommend Authors to revise recent reviews summarising the current NS scenario in mice models and human disease. Furthermore, although Authors have now mentioned genes involved in GBM components regulation, they did not report them in Figure 1 nor in Table 1.
Lastly, how Authors excluded articles (from 366 to 80 included in the study)?
Author Response
To Reviewer 1
The manuscript has significantly improved, however I still have some concerns:
- Authors keep understating the role of GBM and glomerular endothelial cells. Few sentences in the introduction section are not sufficient to cover the topic, which is an important and emergent topic in nephrotic syndromes research. As previously suggested, I recommend Authors to revise recent reviews summarising the current NS scenario in mice models and human disease. Furthermore, although Authors have now mentioned genes involved in GBM components regulation, they did not report them in Figure 1 nor in Table 1.
Response
As suggested, we have added descriptions of GBM on page 1 (line 34-38) and those of glomerular endothelial cells on page 1-2 (line 42-47), and cited review articles. We have added LAMA5 and USP40 in Figure 1.
- Lastly, how Authors excluded articles (from 366 to 80 included in the study)?
Response
As suggested, we have added “Specifically, papers that were not connected to proteinuria or nephrotic syndrome, or that did not include a mouse model were eliminated.” in the Methods section (page 2, line 60-62).
Reviewer 2 Report
The article was very well revised, and the authors considered all my suggestions. I like Figure 1, which has been added.
Thank you for changing the title; currently, it corresponds to the content of the work. The article is good enough to be accepted and published in Biomedicines.
Author Response
To Reviewer 2
The article was very well revised, and the authors considered all my suggestions. I like Figure 1, which has been added.
Thank you for changing the title; currently, it corresponds to the content of the work. The article is good enough to be accepted and published in Biomedicines.
Response
Thank you for your kind comments.
Round 3
Reviewer 1 Report
The manuscript has significantly improved. Although I do not personally think the submitted paper summarises significant novel findings, as the other Reviewer said, I do believe it has reached a sufficient quality to be published in Biomedicines.
MCD (and not only FGS and DKD) has been associated with endothelial dysfunction too (10.1016/j.ekir.2021.11.037). As previously suggested, in a review article it is important to make the point on all aspects of a disease and all novel findings that need more research..
Author Response
To Reviewer 1
The manuscript has significantly improved. Although I do not personally think the submitted paper summarises significant novel findings, as the other Reviewer said, I do believe it has reached a sufficient quality to be published in Biomedicines.
MCD (and not only FGS and DKD) has been associated with endothelial dysfunction too (10.1016/j.ekir.2021.11.037). As previously suggested, in a review article it is important to make the point on all aspects of a disease and all novel findings that need more research..
Response
As suggested, we have added “The endothelium may play a role in some glomerular diseases because it involves alterations of the systemic and glomerular endothelium and glycocalyx, and their sera directly activate GECs [10].” on page 1-2 (line 45-37) by citing the suggested article.